# Infrared and Visible Image Fusion Method Using Salience Detection and Convolutional Neural Network

**DOI:** 10.3390/s22145430

**Published:** 2022-07-20

**Authors:** Zetian Wang, Fei Wang, Dan Wu, Guowang Gao

**Affiliations:** 1School of Electronic Engineering, Xi’an Shiyou University, Xi’an 710065, China; 20212030412@stumail.xsyu.edu.cn (Z.W.); 200102@xsyu.edu.cn (F.W.); wudan@xsyu.edu.cn (D.W.); 2State Key Laboratory of Advanced Design and Manufacturing for Vehicle Body, Hunan University, Changsha 410082, China

**Keywords:** image fusion, salience detection, convolution neural network

## Abstract

This paper presents an algorithm for infrared and visible image fusion using significance detection and Convolutional Neural Networks with the aim of integrating discriminatory features and improving the overall quality of visual perception. Firstly, a global contrast-based significance detection algorithm is applied to the infrared image, so that salient features can be extracted, highlighting high brightness values and suppressing low brightness values and image noise. Secondly, a special loss function is designed for infrared images to guide the extraction and reconstruction of features in the network, based on the principle of salience detection, while the more mainstream gradient loss is used as the loss function for visible images in the network. Afterwards, a modified residual network is applied to complete the extraction of features and image reconstruction. Extensive qualitative and quantitative experiments have shown that fused images are sharper and contain more information about the scene, and the fused results look more like high-quality visible images. The generalization experiments also demonstrate that the proposed model has the ability to generalize well, independent of the limitations of the sensor. Overall, the algorithm proposed in this paper performs better compared to other state-of-the-art methods.

## 1. Introduction

Due to technical bottlenecks in hardware devices, different sensors can only depict a limited amount of scene information from different perspectives. Therefore, image fusion has become the dominant solution in order to obtain a robust and information-rich image. Image fusion is essentially an image enhancement technique that aims to fuse multiple images captured by different sensors into a single fused image that can provide more information about the scene.

One of the most widespread applications is the fusion of infrared and visible images [1]. Infrared sensors are used to convert the infrared thermal radiation of an object into a visible thermal image (i.e., infrared image), which can reduce the interference of complex backgrounds and effectively highlight target information [2]. However, the image often lacks detailed texture information, layers and dimensions and contains a great deal of impulse noise [3]. Visible light sensors, on the other hand, reflect the wavelength range that the human eye can see, as it were, in the image. Not only does the image contain more structural and detailed information, but it also has a higher spatial resolution, which is more in line with the human visual system [4]. The results are not robust, however, as visible images are more affected by the environment, light and weather. By fusing infrared and visible images, the important information from the source image is complemented, allowing a fuller picture of the scene to be presented on a single fused image with higher target salience and superior visual perception. Consequently, infrared and visible light fusion is widely used in modern military, target detection, video surveillance and other important fields.

The last few decades have seen unprecedented developments in infrared and visible light fusion algorithms. They can be divided into two categories [5,6]: traditional fusion-based methods and deep learning-based fusion methods. Most traditional fusion-based methods rely on hand-crafted image features and hand-designed fusion rules [7]. The fusion of multiple source images is accomplished by extracting manual features that describe different frequencies. The manual design of image features and fusion rules is often more complex and difficult to implement. Deep learning-based fusion methods have evolved over recent years [8]. In a nutshell, the feature maps of an image are first extracted from a pre-trained deep neural network, after which the feature maps are stitched together and reconstructed to obtain the final fused image [9]. Compared to traditional fusion algorithms, deep learning fusion algorithms do not require design features and the training of neural networks, and fused images do not interfere with each other, allowing for better fusion results. However, challenges still exist.

Many end-to-end models [10] mitigate the issue of not being able to define the information required to guide network training by weighting the differences between pixels, structures and gradients. However, when constructing the loss function, different regions of the source image are processed indiscriminately, and a large amount of redundant information will be introduced in the fusion process. For the final fused image, this redundant information will become invalid and affect the quality of the fused image.

To address the above challenges, STDFusionNet [5] defines the desired information as the salient target (humans and machines primarily pay attention to the regions where salient targets are located, such as pedestrians, vehicles and hunkers) of the infrared image and the background texture information of the visible image and introduces a salient target mask to manually annotate the salient targets of the infrared image, resulting in better fusion results. However, with a wide range of infrared and visible image fusion tasks, a prominent target mask does not solve all problems once and for all. In infrared and visible image fusion tasks, people’s vision is more sensitive to salient targets, which are quickly noticed and have a very high salience [11,12]. With significance detection, object representations in images can be processed automatically.

In this paper, based on the principle of saliency detection, the infrared image is divided into salient targets and background information, and a saliency detection operation is performed on the salient targets. Significance detection can also be described in the neurological discipline as an attention mechanism that aims to focus or narrow down the important parts of the seen object scene. The human cerebral cortex is sensitive to the contrast of an image, and therefore the extraction of image salience features can be achieved through image contrast. A global contrast-based image salience detection algorithm is used to suppress frequently occurring redundant information and to extract and annotate salient information. Frequently, most methods describe the thermal radiation of an infrared image in terms of a pixel intensity distribution and a gradient distribution to describe the detailed texture information of visible light. The design of the loss function is also based on this, with the loss function being pixel loss for infrared images and gradient loss for visible light images. Based on this, we have redesigned the loss function of the infrared images based on the saliency detection principle, thus guiding the network to selectively extract and reconstruct effective features. Given the above design, the proposed network achieves better fusion performance.

The main contributions of this paper include the following two aspects:1.This paper divides the image information into change information and redundant information and introduces global contrast-based image significance detection (LC) for change information. In the image pre-processing process, significance detection is performed on the infrared images while suppressing the redundant information in the images.2.Based on the principle of saliency detection, the loss function of infrared images is redesigned to better guide the training of the network and selectively extract effective features.

The rest of the paper is organized as follows. In Section 2, a brief introduction is given to related work on image fusion and the salience detection algorithm. In Section 3, the detail of fusion method is presented. Section 4 demonstrates relevant experimental content, while conclusions are given in Section 5.

## 2. Related Works

### 2.1. Traditional Fusion-Based Methods

Traditional fusion-based methods include multi-scale variations [1], sparse representations [13] and the subspace [1,14]. The multiscale transform decomposes the original image into components of different scales, where each component represents a sub-image of each scale, fuses the sub-images of the corresponding scale according to a given fusion rule and finally uses the corresponding inverse multiscale transform to obtain the fused image. Multiscale transforms [8] can be subdivided into pyramid transforms, wavelet transforms, non-sampled contour transforms, etc. The sparse representation [11] is a super-complete dictionary learned from a large number of natural images and sparsely encoded to obtain sparse representation coefficients so that the source image is sparsely represented by the learned super-complete dictionary. Subspace is the projection of a high-dimensional input image into a low-dimensional space (inter-subspace). For natural images that have more redundant information, low-dimensional subspaces can capture more of the intrinsic structure of the original image, reducing time consumption and memory footprint. Regarding methods such as principal component analysis (PCA), independent component analysis and non-negative matrix factorization (NMF), although the traditional methods achieved good results at the time, there are still some drawbacks. The main issues are as follows: (1) Multiscale transforms require artificially set fusion rules and are poorly generalized, and misrepresentation and noise can cause visual guilt in the fused image; (2) In sparse representations, specific dictionaries are restricted to specific types of images, and generic dictionaries impose a heavy computational burden. In addition, images need to be represented linearly in subspace, and real-life images are not always linear. In general, traditional fusion methods do not fully consider the characteristics of different modal images, and feature extraction using the same method for different modal images may not extract the most effective information. At the same time, simple fusion rules designed manually using the maximum value method and the average value method are very limiting to the fusion results.

### 2.2. Deep Learning-Based Fusion Methods

Compared to traditional fusion methods, the deep learning-based fusion of infrared and visible images includes supervised models, unsupervised/self-supervised models, GAN, saliency models, transformer models, task-driven models, bootstrap filter-assisted models and other models. DenseFuse [12] is generally considered to be the first method to use a deep learning model for fusing infrared and visible images and is a supervised learning model. It uses a pre-trained self-encoder in image decomposition and reconstruction, exploiting the feature extraction capabilities that a conventional layer is best at. It is well adapted for different scene images compared to traditional fusion methods. However, the lack of down-sampling makes it difficult to extract multi-scale features from the source image. FusionGAN [15] is the first model to use GAN to fuse infrared and visible images. Adversarial learning is used to avoid artificially designed fusion rules. As the generators and discriminators continue to learn adverbially, the fused images retain progressively more visible information, and eventually only the generators are retained for image fusion. However, a single mechanism of resistance can easily lead to imbalances in fusion. NestFuse [16] is a multi-scale fusion model, which can also be seen as an upgraded version of DenseFuse as its encoder structure is not different from DenseFuse. The difference is that the feature maps output from multiple convolutional layers are all fused, providing some protection against the loss of detail due to down sampling. However, attention is still impossible to learn for the fusion mechanism. VIF-Net [17] is a self-supervised model. It looks more at the temperature salience information of the infrared image, uses the pixel magnitude to rate how much information is in each region and takes more information regions into account when calculating losses. This provides insight into how to measure how much information is in each modality. The pixel magnitude, however, may not fully describe the unique information of the image. The classification saliency-based fusion method (CSF) [18] is a fusion method based on salience classification. Unlike STDFusionNet [5], mentioned in the previous section, which selects salience regions within an image, CSF is evaluated based on the interoperability and importance of the output feature maps of multiple conventional layers. This importance-oriented fusion rule helps to retain valuable feature maps, and the resulting salience information can then be used directly as a weighting factor for fusion. CF-Net [13] is an end-to-end model that takes the super-resolution and multi-exposure fusion tasks and performs the processing with one network. Super-resolution facilitates the accuracy of target detection, so the article processes the multi-exposure task and super-resolution together with good results. This task-driven fusion model, which can also be called a multi-task image fusion model, is very much in line with the future trend of image fusion. Although the multi-exposure image fusion dataset has a ground truth compared to the IR-visible image fusion dataset, a supervised learning model cannot be applied directly.

In general, through the continuous efforts of many scholars, image fusion models have evolved from the initial supervised models or direct application of pre-trained models to self-supervised models and GAN models. To summarize, there are still some open problems that are unsolved in these approaches, which can be generally considered in two aspects: (1) complementary information is not well defined as well as computed and fused for each modality of the image; (2) fused images lose detailed texture information of the main target and do not serve to reduce the cognitive burden on the supervisor.

## 3. Proposed Method

In this section, the specifics of the infrared and visible image fusion method are given.

### 3.1. Problem Formulation

Infrared and visible image fusion is the process of extracting valuable information from two source images and fusing them into one image for presentation. The key to this problem is how to select the most valuable information. Unique to IR images is the magnitude of the pixel value, which reflects the salience of temperature: the higher the temperature, the larger the target pixel value. What is unique to the visible image is the texture detail, with most blurred areas in the infrared image having good detail in the visible image.

In general, infrared images can maintain good temperature significance, and visible images can maintain good detail in texture. Therefore, the key to image fusion is to determine the significance of the infrared image and extract the required information for effective fusion and reconstruction. In other words, the fused image should accurately contain the significant nature of the IR image and the detailed texture of the visible image with as much clarity as possible.

In fusion, we define the fusion process as a regression model, which is represented as
(1)$$I_{f}=Fu\left[f_{1}\left(I_{ir}\right)f_{2}\left(I_{vi}\right)\right]$$
where $I_{ir}$, $I_{vi}i$ and $I_{f}$ is the infrared image, the visible image, and the fused image, respectively, $f_{1}\left(I_{ir}\right)$ and $f_{2}\left(I_{vi}\right)$ are the feature extraction part of the infrared image and the visible image, respectively, and $F_{u}$ denotes the process of feature fusion. Our ultimate aim is to calculate the three mapping functions $f_{1}\left(I_{ir}\right)$, $f_{2}\left(I_{vi}\right)$ and $F_{u}$ by means of a regression model.

### 3.2. Network Architecture

The LCFusion (LCF) architecture is illustrated in Figure 1 and consists of three main components: global contrast-based image saliency detection [19], the feature extraction network [5] and the feature fusion network [20].

As shown in Figure 1, the infrared image is input to the feature extraction network after significance detection, while the visible image is input directly into the feature extraction network. The feature extraction network is made up of a common convolution layer and three residual blocks. The convolution layer consists of a 5 × 5 convolution kernel and swish activation function. The residual block is made up of a residual part consisting of three convolution layers and a direct mapping part. All convolution kernels are 1 × 1 in size, except for the second convolution kernel (red frame), which is 3 × 3. It is clear that the same network structure is used for the feature extraction of the infrared and visible images, but given the different characteristics of the infrared and visible images, combined with their respective loss functions, the parameters of the two are trained independently to better extract salient targets and detailed texture information.

Unlike the feature extraction network, the feature fusion network consists of four residual blocks. The residual block is made up of a residual part consisting of three convolution layers and a direct mapping part. All convolution kernels are 1 × 1 in size, except for the second convolution kernel (red frame), which is 3 × 3. In the fourth residual block, a Tanh activation function is used in the last layer to ensure that the range of variation of the fused image is the same as that of the input image. The input to the feature fusion network is a stitching of the infrared image features with the visible image features in the channel dimension. To avoid information loss during the fusion process, the padding is set to SAME in all convolution layers of the network, and the fused image is the same size as the source image.

### 3.3. Loss Function

In this section, an explicit definition of the loss function is presented; that is, the significance loss $L_{ir}^{sali}$ for infrared images and the gradient loss $L_{vi}^{grad}$ for visible images. The significance loss expects the fused image to retain salient pixel intensities, while the gradient loss forces the fused image to contain more detailed texture information [3,21]. On the one hand, after the infrared image has passed significance detection, the pixel gap between salient and redundant information is more obvious, the edges of the salient target are sharpened, and the redundant information in the background is darker, reducing the interference of image noise to some extent. Therefore, we specifically set a threshold value τ as a criterion for differentiation.

The saliency loss function is defined as
(2)$$L_{ir}^{sali}=\frac{1}{HW}\sum_{i=1}^{H}\sum_{j=1}^{W}\|\frac{Sign\left(I_{i,j}-\tau \right)+1}{2}\times I_{i,j}{\|}_{2}^{2}$$
where *H* and *W* are the height and width of the image, respectively, and ${\left\|•\right\|}_{2}$ stands for the $l_{2}-norm$. τ is the threshold value used to filter the highlighted areas. Sign() is the Sign function. The subscripts *i* and *j* mean the pixels are in row *i* and column *j*.

On the other hand, gradient loss is introduced to tighten the constraints on the network, forcing the fused image to have sharper and more detailed texture information, with prominent target edges that are sharpened and clearer. Somewhat similar to the salience loss definition, the visible gradient loss is defined as
(3)$$L_{vi}^{grad}=\frac{1}{HW}\sum_{i=1}^{H}\sum_{j=1}^{W}\|\nabla I_{f}-\nabla I_{vi}{\|}_{2}^{2}$$
where ∇ denotes the gradient operator; in this essay, we employ the Sobel operator to compute the gradient of an image.

Since the two types of loss functions are not of the same order of magnitude, in order to achieve an approximate de-weighing between the infrared and visible image features, a hyper-parameter is needed to weigh the effects between them. Thus, the final desired loss function is defined as follows: (4)$$L_{total}={\lambda \times L}_{ir}^{sali}+L_{vi}^{grad}$$
where λ is the hyperparameter that controls the loss balance in different regions.

## 4. Experimental Results and Analysis

In this section, the data set, training details and evaluation metrics are presented. This is followed by experiments on a public dataset and a qualitative and quantitative comparison with six representative fusion methods. These include LP [22], GTF [23], MSVD [24], Nestfuse [16], VIF [17] and STD [5]. All six methods are implemented based on publicly available code, and parameters are set according to the experiments in the original paper. In addition, the validity of the specific design is verified through generalization experiments.

### 4.1. Experimental Results

#### 4.1.1. Implementation Parameters

In our earlier reading of the literature, a large number of papers [2,3,4,5,9,11,12,13,14,15,16,19,25,26,27,28,29,30] used the TNO dataset for the training and testing of the model. Therefore, the same TNO dataset is used in this paper. The TNO dataset [31] is a classical dataset for infrared and visible image fusion, which includes different military-relevant scenes registered with different waveband camera systems (including Athena, DHV, FEL and TRICLOBS). We collected 88 pairs of infrared and visible images containing different salient targets from the TNO image dataset. We adopted the Adam optimizer to train the model, the batch size was set to 32, and the learning rate is set to 5 × 10${}^{-4}$. Our experiments were implemented on TensorFlow [21] and trained on a PC with an AMD Ryzen 7 5800H CPU and a NVIDIA RTX3060 GPU. The source code for the proposed method is publicly available [32].

#### 4.1.2. Evaluation Indicators

Currently, the mainstream assessment of fusion performance is divided into qualitative and quantitative evaluations. Qualitative evaluation is also subjective and relies heavily on the human visual system to perceive whether the fused result contains salient targets and detailed texture information. However, subjective evaluation is easily influenced by human factors such as visual acuity, subjective preference and personal emotion. Therefore, fusion performance analysis based on quantitative evaluation is essential and complements the subjective evaluation. Drawing on the evaluation metrics found in most fusion papers [3,4,5,11,12,14,15,16,25,26,27], four mainstream evaluation metrics have been selected for this paper, including the Peak Signal Noise Ratio (PSNR) [26], Structure Similarity Index (SSIM) [4,12,15,16,27], Spatial Frequency (SF) [3,5,11,15,25] and Mutual Information (MI) [3,4,5,14,16]. They are defined as follows: (5)$$PSNR=10\times \log_{10}\left(\frac{z^{2}}{MSE}\right)$$
where *z* represents the difference between the maximum and minimum greyscale values of the ideal reference image, usually 255. MSE represents the mean square error between the source image and the fused image.
(6)$$SSIM\left(x,y\right)=\frac{\left(2\mu_{x}\mu_{y}+c_{1}\right)\left(2\sigma_{xy}+c_{2}\right)}{\left(\mu_{x}^{2}+\mu_{y}^{2}+c_{1}\right)\left(\sigma_{x}^{2}+\sigma_{y}^{2}+c_{2}\right)}$$
where $c_{1}$, $c_{2}$ are constants. $\mu_{x}$ is the mean of *x*, $\mu_{y}$ is the mean of *y*, $\delta_{x}^{2}$ is the variance of *x*, $\delta_{y}^{2}$ is the variance of *y*, and $\delta_{xy}$ is the covariance of *x* and *y*. SSIM takes values in the range [0,1], with larger values indicating less image disturbance.
(7)$$SF=\sqrt{{RF}^{2}+{CF}^{2}}$$

Among them,
(8)$$RF=\sqrt{\frac{1}{MN}\sum_{i=1}^{M}\sum_{j=1}^{N}|H\left(i,j\right)-H\left(i,j-1\right){|}^{2}}$$
(9)$$CF=\sqrt{\frac{1}{MN}\sum_{i=1}^{M}\sum_{j=1}^{N}|H\left(i,j\right)-H\left(i,j-1\right){|}^{2}}$$
(10)$$MI=\sum_{i_{A}\in I_{F}}\sum_{i_{F}\in I_{F}}p\left(i_{A},i_{F}\right)\log_{2}\frac{p(i_{A},i_{F})}{p\left(i_{A}\right)p\left(i_{F}\right)}+\sum_{i_{B}\in I_{F}}\sum_{i_{F}\in I_{F}}p\left(i_{B},i_{F}\right)\log_{2}\frac{p(i_{B},i_{F})}{p\left(i_{B}\right)p\left(i_{F}\right)}$$
where $p(i,i_{F})$ defines the joint probability distribution of $i_{A}$ and $i_{B}$. p(i) is the marginal probability distribution.

### 4.2. Comparative with State-of-the-Arts

#### 4.2.1. Qualitative Evaluation

The fused images obtained by existing fusion methods and our fusion method are shown in Figure 2, Figure 3 and Figure 4. In order to more visually observe the differences in fusion performance between the various algorithms, we selected a prominent region (i.e., the red box) in each fusion image and zoomed in and placed it at the bottom right for clear comparison [33,34].

As shown in Figure 2, the edges of the LP are lost to some extent, and the tails become blurred. The GTF and VIF retain the salience targets of the infrared image but lose the detailed texture information of the visible image due to the smoothing of the texture by the GTF. The VIF similarly loses the detailed texture information and is contaminated with noise from the visible image. The MSVD, Nestfuse and STD results are relatively similar, with the MSVD producing artifacts and poorly defined targets, and the Nestfuse and STD being more heavily contaminated by noise. In contrast, the LCF has sharper target edges, more detailed texture information and clearer details on the helicopter fuselage, making the whole image appear sharper.

In Figure 3, the little man is clearer, and the image is better overall. In Figure 4, our approach allows the body and wheels to be more clearly distinguished. Comparing to Figure 2 and Figure 4, there is more detailed background information. Compared to other fusion results, our method not only effectively highlights the salient targets in the scene but also does a good job of retaining some of the detailed texture information in the background areas. Specifically, the bushes in the GTF, MSVD and VIF in Figure 3 appear blurred, and the edges overlap each other and the background. Nestfuse, STD and LCF reproduce the bushes well in the visible image and distinguish the bushes from their surroundings. Furthermore, in Figure 4, only STD and LCF reproduce the bushes on the distant hillside very well. The comparison reveals that LCF performs better both in highlighting salient targets and in retaining detailed texture information. The infrared images processed by the significance detection mechanism are involved in the fusion, which leads to better results.

#### 4.2.2. Quantitative Evaluation

A quantitative comparison of the seven fusion results was chosen to avoid the interference of subjective human factors [35,36,37]. All fused images are based on publicly available code, and all data in the tables are from our own measurements of the fused images. In general, as a single indicator cannot objectively measure the quality of fusion, four reliable indicators are chosen in this paper to assess the different methods. As shown in Table 1, the proposed method was optimal in three metrics—SSIM, SF and MI—with improvements of 6%, 12.65% and 38.99%, respectively, over the second place. This indicates that the fusion result of LCF not only has enough gradient information but also transfers more information from the source image to the fused image, and the fused image has a better visual effect. Although not a great performer in terms of PSNR metrics, it is still in second place. In contrast, LCF fusion results have an excellent ability to retain salient targets and detailed texture information without significant distortion or artifacts.

### 4.3. Generalization Experiment

In the previous subsection of the comparison experiment, all training and test sets were derived from the TNO dataset and therefore had the same data distribution. In practical applications, however, the test images may come from a different data distribution than the one used for training. These unseen data may differ from the training data in terms of viewpoint, size scale, scene configuration, camera attributes, etc. Therefore, the generalization ability of the network is an important basis for evaluating the fusion model.

To evaluate the generalization capability of the fused model, pairs of images from the RoadScene dataset [20] were selected for testing. Unlike the TNO dataset, the visible images contained in the RoadScene dataset are in color and have a much richer road, vehicle and pedestrian scene. During the fusion process, the source image was not processed in any way and was fed directly into the network for fusion.

#### 4.3.1. Qualitative Evaluation

The qualitative results are intuitively shown in Figure 5. The highlighted areas (i.e., red, green and blue boxes) were also selected for annotation. From top to bottom, the visible image, the infrared image, LP, GTF, MSVD, Nestfuse, VIF, STD and LCF are shown. As can be seen from the fusion results, the texture structure of the visible image is almost completely preserved, and the edges of salient targets in the infrared image are sharpened more. For other fusion methods, the thermal infrared information of the salient targets in the infrared image is well preserved, but the detailed texture information of the salient targets themselves is lost. In column a of Figure 5, the white lining showing behind the figure in the green box and the logo on the back of the figure’s clothing in the red box have become very blurred and difficult to distinguish. In addition, the sky in the fused image is heavily contaminated with thermal information, from which it is difficult to estimate the current time and weather conditions, which is fatal for road scenes. Furthermore, other methods are poor at preserving detailed texture information, such as insulators on poles, street lights and bicycles and text on walls. In contrast, LCF fused images are visually better, and the whole image is much clearer.

#### 4.3.2. Quantitative Evaluation

As with the quantitative evaluation of the comparison trials, all fusion images for the generalization experiments were obtained based on publicly available code, and all data in the tables are derived from our own measurements of the fusion images. The image pairs selected from the RoadScene dataset were objectively evaluated, and the performance of the different fusion methods on the four metrics is shown in Table 2. It can be seen that LCF continues to perform well. It still outperformed the other methods in both SSIM and MI indicators, improving by 17% and 50.22% compared to the second place. As for the PSNR and SF indicators, LCF was only behind by a small margin.

Overall, both qualitative and quantitative results show that LCF has good generalization capabilities and that fusion capabilities are relatively unaffected by image sensor characteristics.

## 5. Discussion

### 5.1. Significance Testing

It is generally accepted that a good significance detection model should satisfy at least three criteria: (1) the actual significant region is not easily lost; (2) the likelihood of incorrectly labeling the background as a significant region is low; (3) the model should detect the significant region quickly. Applying significance detection to infrared images gives good results compared to visible images where the target is not well defined. To verify the impact of significance detection on the final fused images, two models were trained on the TNO dataset. The network optimization was guided based on whether or not significance detection was applied to the infrared images. Specifically, one set of models fed the infrared images out of the fusion network after performing salience detection, while the other set fed the infrared images directly into the fusion network.

As shown in Figure 6, the comparison reveals that the contrast between the highlighted target and the background is more pronounced in the infrared image after significance detection. Processing the unwanted background closer to full black reduces the interference of the noise in the infrared image itself. Secondly, by comparing the fused images, the fusion results of feeding the infrared images directly into the fusion network are unsatisfactory, containing noise that has not been preconditioned and visually unsatisfactory. Conversely, images that have undergone LCF fusion have better contrast and clarity, sharpened object edges and sharper target textures, with better visual effects, both in terms of background texture information and salient targets.

### 5.2. Loss Function

The information unique to infrared images is pixel magnitude information, reflecting temperature salience, where the higher the temperature, the larger the target pixel. For most fusion models, the loss function for infrared images use MSE and L2−norm as a way of maintaining target significance information. In this paper, due to the desire to retain salient pixels and rich gradients, significance detection is performed on infrared images, and we designed a special loss function for this purpose. To verify the effectiveness of the designed loss function, two comparison models were also trained on the TNO dataset. One applied the conventional MSE as the loss function, and one applied the specially designed loss function.

As shown in Figure 7, the contrast between the fused images is quite obvious. The fused image under the conventional loss function constraint can be described as a disaster in terms of the fusion effect. Neither are the salient targets well highlighted, nor is the background well treated, and even noise is added and the fused image becomes distorted. In contrast, the integration of the proposed method is even better.

### 5.3. Different Parameters

The previous section qualitatively compared the impacts of different loss functions on the final fusion results. Figure 8 shows a comparison of the network’s ability to converge in early training. As can be seen in Figure 8, the LCF loss function for infrared images achieves faster convergence than the common MSE loss function for infrared images, which greatly reduces the time loss in the training phase and justifies our loss function design.

In order to ensure the best performance of the fusion framework, the choice of the different parameters is discussed next. These include the choice of patch size for the input network during training, the choice of hyperparameters τ in the loss function for the infrared images and the comparison of the convergence speed of the network under different loss functions. As shown in Figure 9a shows the comparison of the effect of different patch sizes on the final fusion results, and Figure 9b shows the comparison of the effect of different hyper-parameters τ in the loss function of the infrared images on the final fusion results. In trained dataset, the minimum resolution of the images is 280×280. Therefore, for patch selection, 16×16, 32×32 and 64×64 were selected for comparative study, and we selected the evaluation metric SF for the quantitative comparison of the final fused images. It can be seen from Figure 9a that the 16×16 size patches (blue) correspond to higher evaluation metrics for the fused images. Therefore, the patches of size 16×16 were finally chosen.

The hyperparameters τ are also selected and compared in order to achieve the best performance of the designed loss function in the training of the network. As shown in Figure 9b, the hyperparameters τ were set to 0.2, 0.3, 0.4, 0.5 and 0.6 to weigh and compare the differences between them. When τ = 0.3, the SF metrics of the fused images are even better. Therefore, based on the consideration of quantitative indicators, we believe that the best fusion effect can be obtained when τ = 0.3.

### 5.4. Add Noise

Given the more or less good quality of the test dataset, some attempts were made to explore the capabilities of our algorithm in difficult conditions by applying the same method to source images taken in bad weather or in low light conditions. Thus, a set of comparison experiments is added to this subsection. In the experiments, Gaussian noise and pepper noise, which are difficult for the human eye to distinguish, are artificially added to the visible images. The fusion was then performed, and several methods were selected for comparison.As shown in Figure 10.

The MSVD method was the best in terms of noise removal, and the saliency target was very noticeable. However, this fusion method lost the detailed texture of the salient targets themselves. We would like to obtain more comprehensive information about the scene, containing not only the salient target and background details but also the detailed texture of the salient target, which would reduce the cognitive burden on the supervisor. Among the three similar fusion results, NestFuse, STD and LCF, this paper’s fusion result has the biggest advantage of having a clearer detail texture of the salient target. In comparison, the fusion result of this paper contains the least noise among the three fusion results. It also demonstrates that the proposed algorithm in this paper has some noise removal ability under difficult conditions.

Of course, this paper has only made some small attempts in this subsection, and future research will be carried out on images (images with poor quality, a large amount of noise and loss of sharpness) acquired under more extreme weather conditions (e.g., foggy days or low light conditions) Examples include image pre-processing operations, image fusion algorithms, etc.

## 6. Conclusions

In this paper, we propose an infrared and visible image fusion network based on significance detection. Firstly, a global contrast-based significance detection algorithm extracts the salient features of the infrared image, highlighting high intensity values and suppressing low intensity values and image noise. Secondly, a special loss function is designed for infrared images to guide the extraction and reconstruction of features in the network, based on the principle of prominence detection, while the more mainstream gradient loss is used as the loss function for visible images in the network. Extensive experiments, qualitative and quantitative evaluations have shown that our fused images are sharper and more visually appealing. Our approach also has excellent generalization performance and can be applied to other datasets or other image fusion tasks.

## Figures and Tables

**Figure 1 sensors-22-05430-f001:**
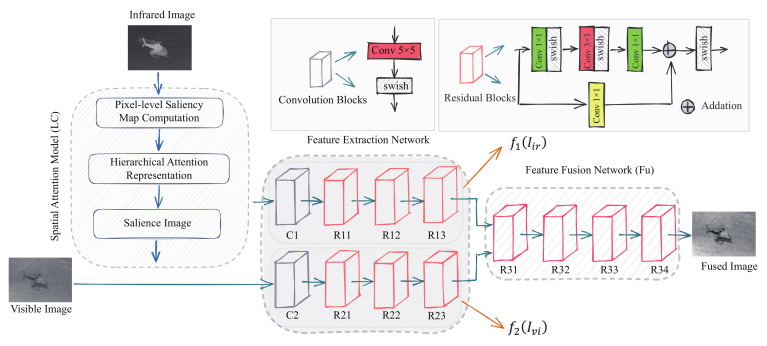
Architecture of the proposed infrared and visible image fusion network.

**Figure 2 sensors-22-05430-f002:**
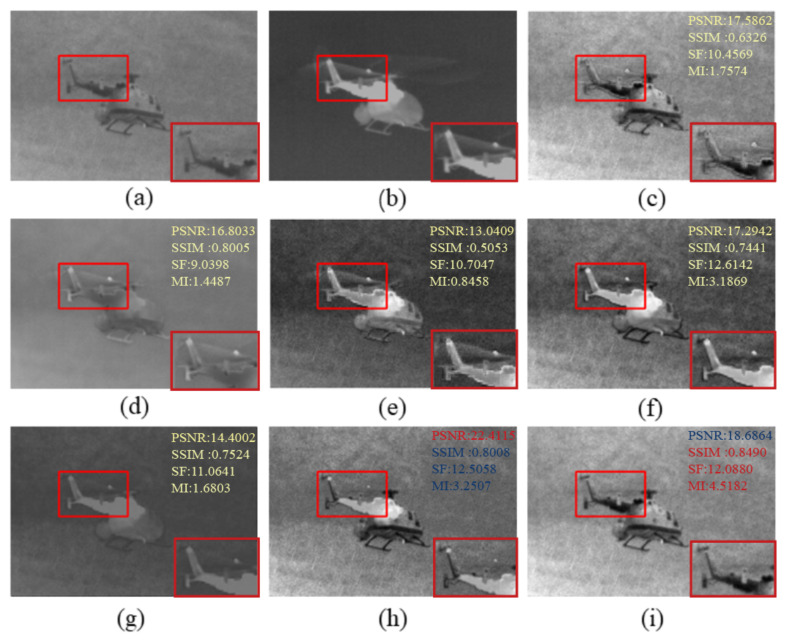
Qualitative comparison of LCF with six state-of-the-art methods on a helicopter. For a clear comparison, we select a salient region (i.e., the red box) in each image and enlarge it in the bottom right corner. In addition, the evaluation indicators are displayed on the top right, with red representing the best and blue the second best. (**a**) Visible image; (**b**) Infrared image; (**c**) LP; (**d**) GTF; (**e**) MSVD; (**f**) Nestfuse; (**g**) VIF; (**h**) STD; (**i**) LCF.

**Figure 3 sensors-22-05430-f003:**
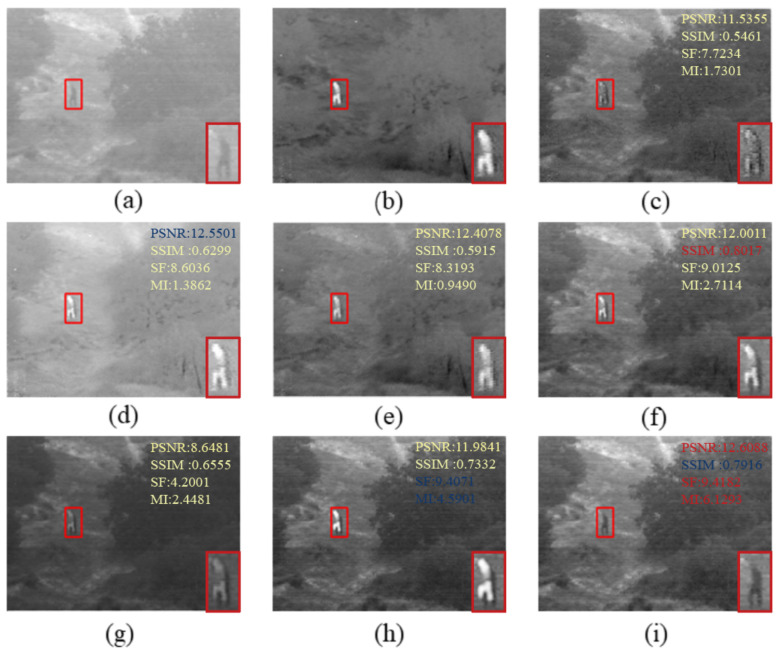
Qualitative comparison of LCF with six state-of-the-art methods on people. For a clear comparison, we select a salient region (i.e., the red box) in each image and enlarge it in the bottom right corner. In addition, the evaluation indicators are displayed on the top right, with red representing the best and blue the second best. (**a**) Visible image; (**b**) Infrared image; (**c**) LP; (**d**) GTF; (**e**) MSVD; (**f**) Nestfuse; (**g**) VIF; (**h**) STD; (**i**) LCF.

**Figure 4 sensors-22-05430-f004:**
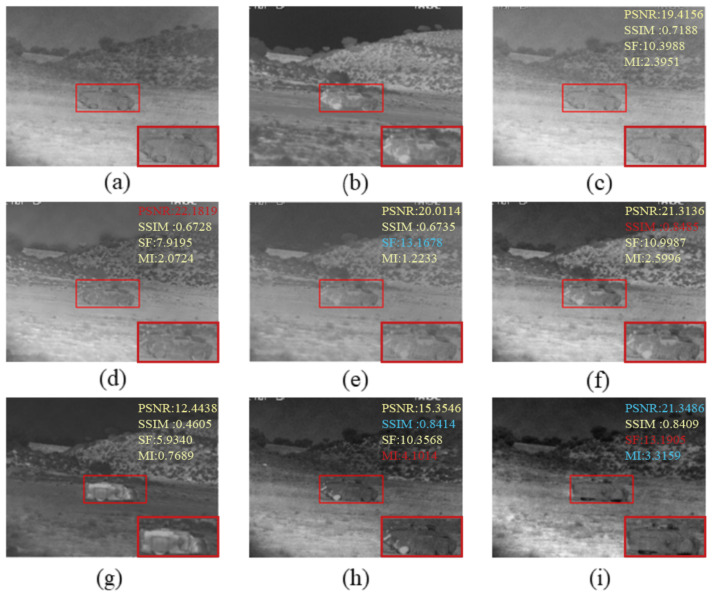
Qualitative comparison of LCF with six state-of-the-art methods on armored. For a clear comparison, we select a salient region (i.e., the red box) in each image and enlarge it in the bottom right corner. In addition, the evaluation indicators are displayed on the top right, with red representing the best and blue the second best. (**a**) Visible image; (**b**) Infrared image; (**c**) LP; (**d**) GTF; (**e**) MSVD; (**f**) Nestfuse; (**g**) VIF; (**h**) STD; (**i**) LCF.

**Figure 5 sensors-22-05430-f005:**
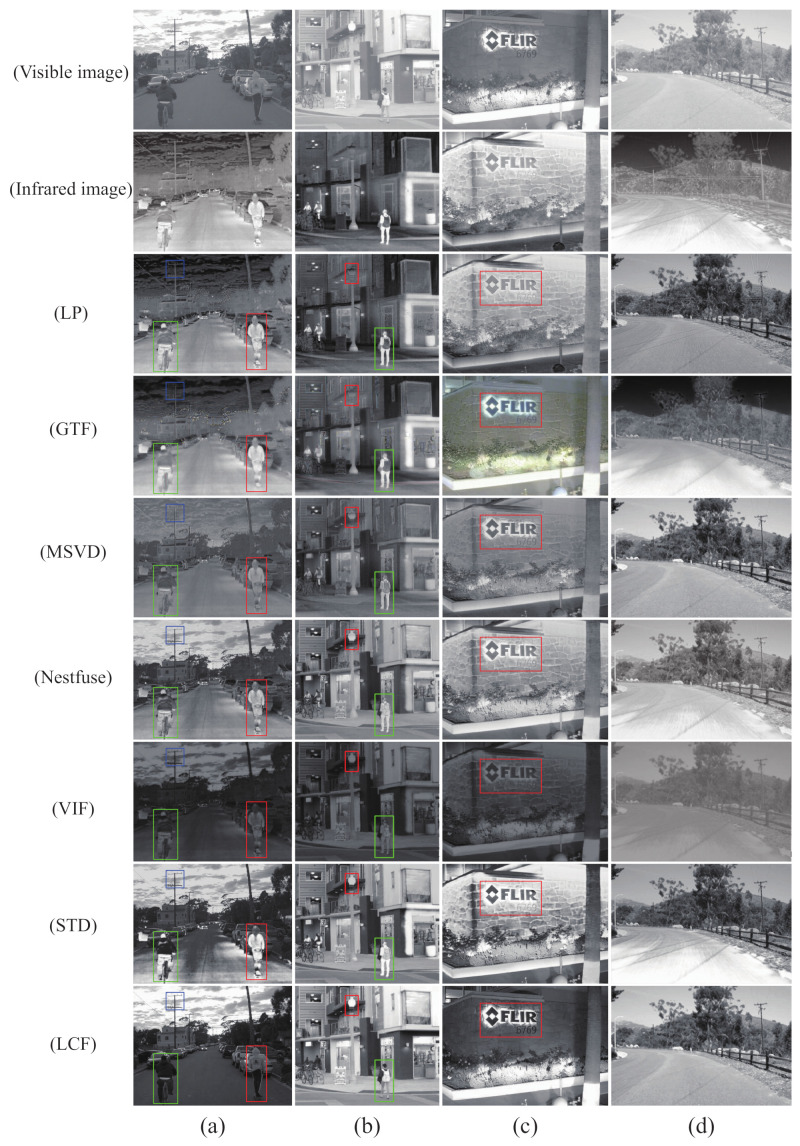
Qualitative comparison of LCF with six state-of-the-art methods on a street. Each row is an image from the RoadScene dataset (**a**–**d**). For a clear comparison, we select a salient region (i.e., the red box, the green box, the blue box) in each image and enlarge it in the bottom right corner. From top to bottom, each row is as follows: Visible image. Infrared image. LP. GTF. MSVD. Nestfuse. VIF. STD. LCF.

**Figure 6 sensors-22-05430-f006:**
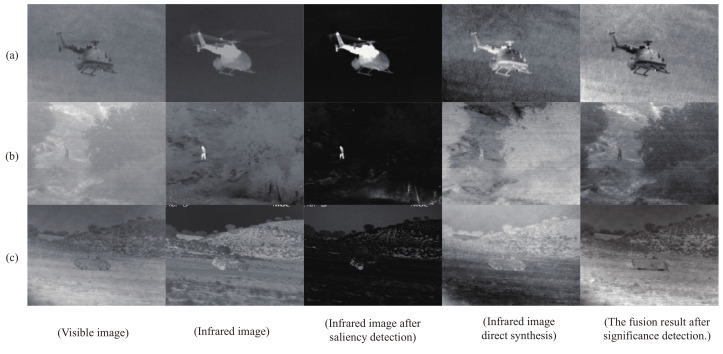
Comparison of images regarding significance detection. Each row is an image from the TNO dataset (**a**–**c**). From left to right, each row is as follows: Visible image. Infrared image. Infrared image after saliency detection. Infrared image direct synthesis. The fusion result after significance detection.

**Figure 7 sensors-22-05430-f007:**
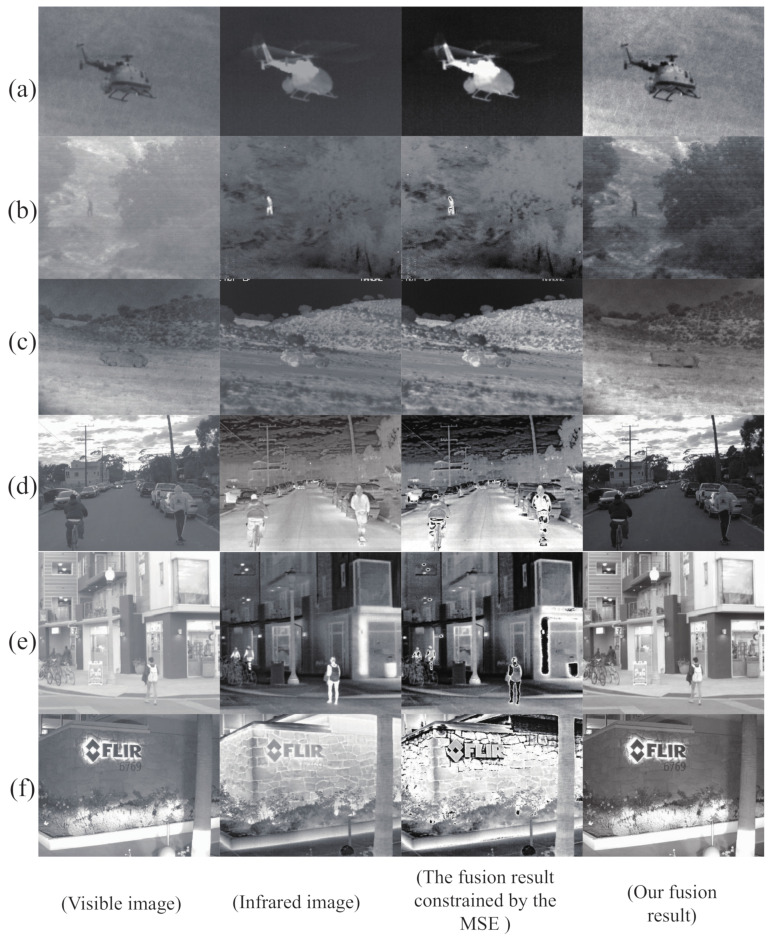
Comparison of images on different loss functions. The panels of (**a**–**f**) are images derived from the TNO dataset and the Roadscene dataset. From left to right, each row is as follows: Visible image. Infrared image. The fusion result constrained by the MSE. Our fusion result.

**Figure 8 sensors-22-05430-f008:**
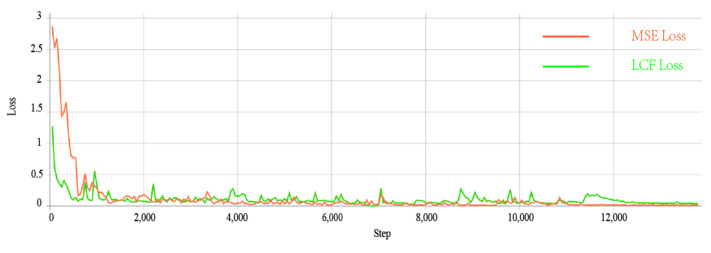
Comparison on different loss functions.

**Figure 9 sensors-22-05430-f009:**
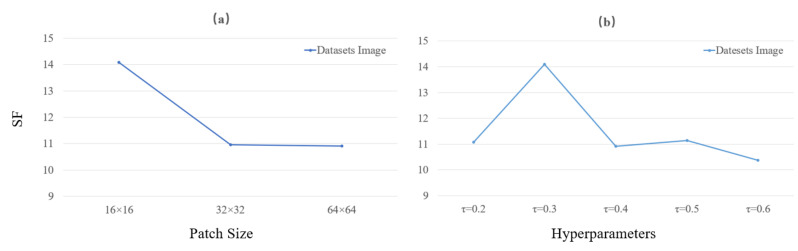
Performance with different parameters. (**a**) A comparison of different patches; (**b**) A comparison of different hyperparameters τ.

**Figure 10 sensors-22-05430-f010:**
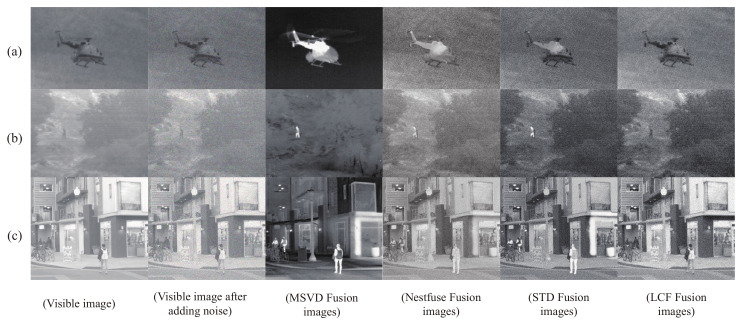
Comparison of fusion results after adding noise. The pabels of (**a**–**c**) are three different images respectively. From left to right, each row is as follows: Visible image. visible image after adding noise. Fused image of MSVD. Fused image of NestFuse. Fused image of STD. Fused image of LCF.

**Table 1 sensors-22-05430-t001:** Quantitative comparison of different methods for the fusion of TNO dataset. Red indicates the best result, and blue represents the second best result.

Methods	PSNR	SSIM	SF	MI
LP	17.53 ± 1.82	0.65 ± 0.08	10.43 ± 2.73	1.75 ± 0.02
GTF	16.63 ± 4.25	0.86 ± 0.17	9.03 ± 0.43	1.48 ± 0.10
MSVD	13.14 ± 0.63	0.56 ± 0.08	10.72 ± 2.38	0.87 ± 0.10
Nestfuse	17.24 ± 5.29	0.78 ± 0.05	12.67 ± 3.61	3.18 ± 0.47
VIF	14.40 ± 5.75	0.72 ± 0.09	11.08 ± 6.86	1.68 ± 0.76
STD	22.43 ± 9.42	0.82 ± 0.06	12.53 ± 3.09	3.25 ± 1.33
LCF	18.98 ± 7.07	0.84 ± 0.05	14.07 ± 4.66	4.51 ± 1.61

**Table 2 sensors-22-05430-t002:** Quantitative comparisons of different methods for the fusion of TNO dataset. Red indicates the best result, and blue represents the second best result.

Methods	PSNR	SSIM	SF	MI
LP	7.61 ± 0.54	0.34 ± 0.11	11.62 ± 7.91	1.23 ± 0.71
GTF	7.63 ± 0.03	0.55 ± 0.07	9.67 ± 2.18	1.31 ± 0.71
MSVD	12.17 ± 3.97	0.58 ± 0.14	14.16 ± 4.36	0.79 ± 0.12
Nestfuse	15.86 ± 3.53	0.69 ± 0.11	13.44 ± 3.15	1.64 ± 0.57
VIF	9.57 ± 2.87	0.54 ± 0.01	7.88 ± 1.37	1.56 ± 0.87
STD	13.96 ± 1.43	0.72 ± 0.06	16.54 ± 3.59	2.84 ± 1.16
LCF	15.38 ± 0.04	0.84 ± 0.03	15.43 ± 4.71	5.77 ± 0.25

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
