# Peer review of "Infrared and Visible Image Fusion Method Using Salience Detection and Convolutional Neural Network"

_sensors, 2022, doi:10.3390/s22145430_

Round 1
Reviewer 1 Report
The reviewed manuscript presents a fusion method for obtaining a synergetic image using two images obtained at different wavelengths, namely, in the infrared and in visible bands.
The resulting method has a lot of applications in the systems of machine vision, the methodology presented in the article is sound, and the paper is well structured and written (except for some minor issues listed in “Technical corrections” section). It is especially interesting that the authors have implemented the other published methods to the same scenes and compared the results of different approaches applied in the same setup to the same input data.
I believe it can be published in the journal if the authors address the following issues and make the corrections listed below
General comments:
Methodologically, I see only one issue in this manuscript – the test datasets are more or less of a good quality and I would say that even a single, visible data set could be used in the object recognition chain after pumping up the contrast and applying something like “unsharp mask”. At the same time, it would be interesting to apply the same methodology to pictures taken in foggy or low-light conditions when the noise is high. I believe, the images of this kind do exist or one can just add a standard noise to the existing images to make them less recognizable for the human eye. I do not ask to redo the whole study, but a separate section or an appendix demonstrating the capabilities of the algorithm in difficult conditions would be welcome.
Specific comments:
Line 59: it would be good to have a quantitative definition of a “salient feature” here. Which feature is called “salient”?
Line 367, Fig. 5: From my point of view, STD outperforms LCF in scenes 1, 2, 3 and maybe even 4. Even though the daytime colors are changed, the features are better detectable in the 8th row than in the 9th one. Please, comment.
Line 404, Fig. 7: again, I can’t say that the last (the right-hand side) column is better than the 3rd one in the 2nd, 4th, and 5th rows.
Technical corrections:
Line 65: language issue, I didn’t get what you mean by “divide the information”, it doesn’t match with the rest of the sentence. Please, reformulate
Line 68: I have the same problem with “performed on the change information” part of the sentence. Please, reformulate.
Lines 83, 84: “divides”, “introduces”
Line 94: I do not get how this sentence is linked with the rest of the section.
Lines 125−126: is the enumeration needed here?
Lines 179−180: the goal is not well explained. I would say that the goal is to make a synergy of two images and to obtain something that is bigger than a simple sum of its components.
Lines 182−186 and elsewhere: the word “information” is overused and overloaded. Please, rewrite in a more precise way.
Line 192: I’d just write “We define the fusion process as…”
Line 196: Please, add an explanation to Fu – is it an operator or function or what?
Line 226: what are the units and meaning of these loss values: 999 and 666 ?
Line 269: “mothed” -> “method”
Line 325 and elsewhere in all the tables: it doesn’t make sense to provide the data with this accuracy. 17.52+/-1.83 or 0.65+/-0.09 would be more than sufficient.
Line 382: “shown”
Figures 5, 6, 7 – please, add the column and row numbers
Lines 391–395 and 414–417: these lines repeat the figure captions, I’d skip them and just describe the figures.
Line 432: I’d just write “Fig9a shows, … Fig. 9b shows”.
Lines 436, 438, and 439: I believe, the dimensions are given here, so the numbers should be separated by “x” symbol: 16x16 and so on.
Line 446: is it a good term and good wording? I mean, “an attention mechanism”
Author Response
The authors thank the editor and all reviewers for the constructive comments, and have very carefully followed and cleaned up every raised issue. We provide below a detailed account on the changes that we have made in response to comments that the editor and reviewers have provided. The corresponding changes in the R1 version are in red color. Apart from addressing these concerns from the editor and reviewers, we have also further polished the presentation, and the changed parts are underlined in blue in the R1 version.Please see the attachment.

Reviewer 2 Report
This article reports an algorithm for the fusion of infrared and visible images. Experimental results are being presented and analyzed in comparison to the state of the art. The manuscript can only be accepted for publication after addressing the comments below.
· Page 3, line 94: There is text containing some references without any purpose. It seems that it was a comment which has not been removed from the manuscript before submission.
· Figure 1: Text is not readable for the blocks located in the upper right corner of the figure.
· Page 7, Line 270: Provide a full link in the list of references at the end of the article and refer to it here in the text.
- Figure 8: Difficult to distinguish two plots in orange and red colours. Please use different colours to be easily distinguished.
· The article seems to be in an early draft stage. There are many problems found in the captions of the figures and the tables (e.g. Table 1). There are many grammatical mistakes found in the text. So, please improve the text.

Author Response

(The authors gave the same response as above.)

Reviewer 3 Report
This paper proposes an algorithm for infrared and visible image fusion using saliency detection and convolutional neural networks and conducts rich comparative experiments with some innovation, but there are still some areas for improvement.
1. The paper still needs to be double-checked to avoid appearances such as line 94 "[1]. Now citing a book reference [2,3] or other reference types [4-6]." Such misdescriptions.
2. The description in the “Proposed Method” section of the paper is not clear enough. For example, the C22 mentioned in line 209 of the paper is not found in figure1, and needs to be further modified.
3. The presentation of experimental results in the paper also needs to be neater, e.g., the method and result data should be aligned in Table 1.
Author Response

(The authors gave the same response as above.)

Round 2
Reviewer 2 Report
All my concerns have already been addressed, so I recommend this manuscript for publication.
Reviewer 3 Report
The paper can be accepted in present form.
This manuscript is a resubmission of an earlier submission. The following is a list of the peer review reports and author responses from that submission.